# Restaging Transurethral Resection of Bladder Tumours after BCG Immunotherapy Induction in Patients with T1 Non-Muscle-Invasive Bladder Cancer Might not Be Associated with Oncologic Benefit

**DOI:** 10.3390/jcm9103306

**Published:** 2020-10-15

**Authors:** Wojciech Krajewski, Marco Moschini, Łukasz Nowak, Sławomir Poletajew, Andrzej Tukiendorf, Luca Afferi, Jeremy Teoh, Tim Muilwijk, Steven Joniau, Alessandro Tafuri, Alessandro Antonelli, Alessandra Gozzo, Andrea Mari, Ettore Di Trapani, Kees Hendricksen, Mario Alvarez-Maestro, Andrea Rodriguez Serrano, Giuseppe Simone, Stefania Zamboni, Claudio Simeone, Maria Cristina Marconi, Riccardo Mastroianni, Guillaume Ploussard, Paweł Rajwa, Ekaterina Laukhtina, Aleksandra Zdrojowy-Wełna, Anna Kołodziej, Andrzej Paradysz, Karl Tully, Joanna Krajewska, Radosław Piszczek, Evanguelos Xylinas, Romuald Zdrojowy

**Affiliations:** 1Department of Urology and Oncologic Urology, Wrocław Medical University, 50-556 Wroclaw, Poland; lllukasz.nowak@gmail.com (Ł.N.); anna.kolodziej@umed.wroc.pl (A.K.); romuald.zdrojowy@umed.wroc.pl (R.Z.); 2Klinik für Urologie, Luzerner Kantonsspital, 6004 Lucerne, Switzerland; marco.moschini87@gmail.com (M.M.); luca.afferi@gmail.com (L.A.); 3Second Department of Urology, Centre of Postgraduate Medical Education, 01-813 Warsaw, Poland; slawomir.poletajew@cmkp.edu.pl; 4Department of Public Health, Wrocław Medical University, 50-556 Wrocław, Poland; andrzej.tukiendorf@gmail.com; 5S.H.Ho Urology, Department of Surgery, Prince of Wales Hospital, The Chinese University of Hong Kong, Hong Kong, China; jeremyteoh@surgery.cuhk.edu.hk; 6Department of Urology, University Hospitals Leuven, 3000 Leuven, Belgium; tim.muilwijk@kuleuven.be (T.M.); steven.joniau@uzleuven.be (S.J.); 7Department of Urology, University of Verona, Azienda Ospedaliera Universitaria Integrata Verona, 37129 Verona, Italy; aletaf@hotmail.it (A.T.); alessandro_antonelli@me.com (A.A.); alessandraguro@gmail.com (A.G.); 8Department of Urology, Careggi Hospital, University of Florence, 50139 Florence, Italy; andreamari08@gmail.com; 9Department of Urology, European Institute of Oncology, 20143 Milan, Italy; ettore.ditrapani@ieo.it; 10Department of Urology, Netherlands Cancer Institute-Antoni Van Leeuwenhoek Hospital, 1066 CX Amsterdam, The Netherlands; k.hendricksen@nki.nl; 11Department of Urology Hospital Universitario La Paz, 28046 Madrid, Spain; malvarezmaestro@hotmail.com (M.A.-M.); andrearodriguez.s@gmail.com (A.R.S.); 12IRCCS “Regina Elena” National Cancer Institute, Department of Urology, 00144 Rome, Italy; puldet@gmail.com (G.S.); riccardo.mastroianni@gmail.com (R.M.); 13Urology Unit, Department of Medical & Surgical Specialties, Radiological Sciences & Public Health, University of Brescia, ASST Spedali Civili of Brescia, Brescia 25123, Italy; stefania.zamboni@libero.it (S.Z.); csimeone@libero.it (C.S.); marconimariac86@gmail.com (M.C.M.); 14Department of Urology, La Croix du Sud Hospital, 31130 Quint Fonsegrives, France; g.ploussard@gmail.com; 15Department of Urology, Medical University of Silesia, 3 Maja Street 13-15, 41-800 Zabrze, Poland; pawelgrajwa@gmail.com (P.R.); parady@poczta.onet.pl (A.P.); 16Institute for Urology and Reproductive Health, Sechenov University, 119146 Moscow, Russia; katyalaukhtina@gmail.com; 17Department of Urology, Comprehensive Cancer Centre, Medical University of Vienna, 1090 Vienna, Austria; 18Department of Endocrinology, Diabetes and Isothope Therapy, Wroclaw Medical University, 50-556 Wroclaw, Poland; aleksandra.zdrojowy-welna@umed.wroc.pl; 19Department of Urology and Neurourology, Marien Hospital Herne, Ruhr-University Bochum, 44625 Herne, Germany; karl.tully@elisabethgruppe.de; 20Department and Clinic of Otolaryngology, Head and Neck Surgery, Wroclaw Medical University, 50-556 Wroclaw, Poland; wojciechowska.joan@gmail.com; 21Department of Urology and Oncologic Urology, Lower Silesian Specialist Hospital, 50-556 Wroclaw, Poland; piszczek.radoslaw@gmail.com; 22Department of Urology, Bichat-Claude Bernard Hospital, Assistance Publique-Hôpitaux de Paris, Paris Descartes University, 75018 Paris, France; evanguelosxylinas@hotmail.com

**Keywords:** bladder cancer, BCG, reTURB

## Abstract

Background and Purpose: The European Association of Urology guidelines recommend restaging transurethral resection of bladder tumours (reTURB) 2–6 weeks after primary TURB. However, in clinical practice some patients undergo a second TURB procedure after Bacillus Calmette-Guérin immunotherapy (BCG)induction. To date, there are no studies comparing post-BCG reTURB with the classic pre-BCG approach. The aim of this study was to assess whether the performance of reTURB after BCG induction in T1HG bladder cancer is related to potential oncological benefits. Materials and Methods: Data from 645 patients with primary T1HG bladder cancer treated between 2001 and 2019 in 12 tertiary care centres were retrospectively reviewed. The study included patients who underwent reTURB before BCG induction (Pre-BCG group: 397 patients; 61.6%) and those who had reTURB performed after BCG induction (Post-BCG group: 248 patients, 38.4%). The decision to perform reTURB before or after BCG induction was according to the surgeon’s discretion, as well as a consideration of local proceedings and protocols. Due to variation in patients’ characteristics, both propensity-score-matched analysis (PSM) and inverse-probability weighting (IPW) were implemented. Results: The five-year recurrence-free survival (RFS) was 64.7% and 69.1% for the Pre- and Post-BCG groups, respectively, and progression-free survival (PFS) was 82.7% and 83.3% for the Pre- and Post-BCG groups, respectively (both: *p* > 0.05). Similarly, neither RFS nor PFS differed significantly for a five-year period or in the whole time of observation after the PSM and IPW matching methods were used. Conclusions: Our results suggest that there might be no difference in recurrence-free survival and progression-free survival rates, regardless of whether patients have reTURB performed before or after BCG induction.

## 1. Introduction

According to the current European Association of Urology (EAU) guidelines, the treatment of patients with high-risk, non-muscle-invasive bladder cancer (NMIBC) should be based on performing transurethral resection of bladder tumour (TURB), followed by 1–3 years of Bacillus Calmette-Guérin (BCG) immunotherapy (induction + maintenance regimen) [1]. One of the significant risks of such a procedure, however, is the presence of one or more residual tumour(s) following initial TURB. This can be especially problematic if the stage of the tumour(s) is underestimated, as is often the case [1]. Because of these disadvantages, a second transurethral resection (reTURB) has been suggested as a potential solution to properly stage the disease, reduce recurrence/progression rates, and improve oncological outcomes following BCG treatment [2,3]. Currently, the EAU guidelines recommend reTURB 2–6 weeks after the primary TURB [1]. However, reTURB timing has not yet been agreed upon, due to the lack of strong evidence behind current recommendations. In the majority of centres in Europe, initial TURB is followed by reTURB before the BCG therapy implementation. However, in some cases, reTURB may be performed after the BCG induction course. This is mainly due to logistical reasons and long waiting lists. In the available literature, no data assessing this sequence of events were found. The aim of this study was to compare the oncological outcomes of patients with T1 high-grade (HG) NMIBC who underwent reTURB according to the EAU guidelines (before BCG therapy) and of those who had reTURB performed after the BCG induction course.

## 2. Materials and Methods

Data sharing from all participating sites used in this observational, retrospective cohort study was approved by an institutional review board. Data from 645 patients from 12 tertiary care centres with primary T1HG NMIBCs, with or without concomitant carcinoma in situ (CIS), who were treated with reTURB and BCG maintenance immunotherapy between 2001 and 2019 were included in the retrospective assessment. The study included patients who underwent the reTURB procedure before BCG induction (Pre-BCG group: 397 patients; 61.6%), as well as those who had reTURB performed after a BCG induction course (5–6 weekly instillations)—Post-BCG group: 248 patients, 38.4% (Figure 1). The decision to perform the reTURB procedure before or after a BCG induction course was determined by the surgeon with the consideration of local protocols. Enhanced visualization modalities (e.g., photodynamic diagnostics/narrow-band imaging) were also used during the procedure, at the urologists’ discretion and based on equipment availability.

All patients were treated with BCG induction and maintenance courses. The BCG instillations were given according to the international guidelines and local protocols at the time, with at least one year of planned maintenance. Every patient included in the study received a minimum of five instillations of induction and at least two maintenance instillations [4]. The BCG strains administered varied between the centres, but mainly TICE, RIVM, Moreau, and Connaught BCG were used.

The sociodemographic data of patients collected comprises age at surgery, gender, and smoking history. The pathological data available included the tumour stage, grade, size, focality, presence of concomitant CIS, and presence of muscularis propria (MP) in the specimen, for both primary and restaging TURB. Data on the immediate, single chemotherapy instillation, lymphovascular invasion (LVI), variant histology (VH), and prostatic involvement of the tumours were not uniformly reported and/or were unreliable, and were therefore not included in this analysis.

Tumours included in this analysis were graded according to the WHO 2004 system. Lesions were staged according to the American Joint Committee on Cancer TNM classification. Specimens were evaluated by dedicated uropathologists at each participating centre; no central assessment was applied. Patients were followed up according to the EAU guidelines at the time.

Concomitant CIS was defined as the coexistence of carcinoma in situ and an exophytic tumour. A recurrence was defined as the reappearance of a tumour of any stage, and the grade was confirmed by TURB and histologic assessment. Residual tumours at reTURB and tumour recurrence in the upper urinary tract were not considered as a recurrence. Progression was defined as tumour relapse at tumour stage T2 or higher in the bladder or stromal invasion of the prostatic urethra, or as distant (e.g., lymph nodes) progression. Patients with T2 lesions at both pre- and post-BCG reTURB were not included in the analysis as they did not qualify for BCG therapy and/or underwent radical therapy.

The primary database included 1431 high-risk NMIBC patients from 12 centres. The exclusion criteria for this study included incomplete major variables data, tumours other than T1HG, recurrent tumours, incomplete primary TURB with evident residual disease, time interval between TURB and reTURB of >90 days in the Pre-BCG group and >180 days in the Post-BCG group, number of BCG instillations <7, follow-up <6 months, and other than a full dose concentration of BCG for a given strain. After the exclusion process, 645 cases were included.

The main aim of the study was to determine whether administration of the induction BCG course before reTURB results in significantly better oncological outcomes, as defined by recurrence-free survival (RFS) and progression-free survival (PFS).

### Statistical Analysis

The results between study groups were compared using the chi^2^ and Mann–Whitney test. The RFS and PFS were estimated with the log-rank method, and Kaplan–Meier curves were plotted. Additionally, Cox regression analyses were performed for both RFS and PFS. Patients without an adverse event or death before the end of the study were removed from the study after the last follow-up. Times to adverse events were calculated by taking the date of primary resection as time zero. Due to differences in baseline patient characteristics in the groups, we used a 1:1 propensity-score-matched analysis (PSM) adjusted for gender, smoking status, age, presence of MP in primary specimen, tumour focality, size, and concomitant CIS incidence [5]. Moreover, to reduce the bias of unweighted estimators and adjust for covariates’ imbalance between treatment groups without losing patients, inverse probability of treatment weighting (IPW) was used, using the same variables as in PSM [6].

Statistical significance was considered as *p* < 0.05. Statistical analyses were performed using STATA 14 (Stata Corp., College Station, TX, USA) and R platform (R project, Vienna, Austria).

## 3. Results

The study included 397 patients with the reTURB procedure performed before BCG induction (Pre-BCG group: mean age 65.9 ± 11.8 y; 321 males) and 248 patients with reTURB performed after BCG induction course (Post-BCG group: mean age 67.2 ± 9.9 y; *p* = 0.074, 217 males, *p* = 0.027). Baseline patient characteristics are presented in Table 1.

The median follow-up period for the Pre-BCG group was 40 months (IQR: 27.6–58.6), and it was 45.4 months (IQR: 23.3–58.0) for the Post-BCG group (*p* > 0.001). Groups did not differ statistically in terms of smoking history, tumour size and focality, the presence of MP in the reTURB specimen, and the BCG instillation number. However, patients in the Pre-BCG group had a higher rate of MP presence in the primary TURB specimen, and more residual cancers were found during reTURB. Also, a borderline difference was observed for concomitant CIS, with more observed in the Post-BCG group.

Recurrence was observed in 128 patients (32.2%) in the Pre-BCG group and in 90 patients (36.6%; *p* = 0.290) in the Post-BCG group. Progression was observed in 94 patients (14.6%) in the Pre-BCG group and in 40 (16.1%; *p* = 0.376) in the Post-BCG group. There were 26 (6.5%) and 12 (4.8%) cancer-specific deaths in the Pre-BCG and Post-BCG groups, respectively (*p* = 0.369).

The five-year RFS and PFS were 64.7% vs. 69.1% and 82.7% vs. 83.3% for the Pre-BCG and Post-BCG groups, respectively (both *p* > 0.05) (Figure 2A,B). Also, no differences were found for the whole observation period, either for RFS (Figure 2C) or for PFS (Figure 2D).

Considering multivariate analyses, reTURB timing was not associated with statistically significant differences for RFS or PFS (Table 2). Positive reTURB increased the risk of recurrence and progression by 2.6- and 3-fold, respectively.

Because of the retrospective and multicentre nature of the study, some disparities in baseline patient characteristics were observed. Therefore, the PSM was performed. The groups were adjusted for gender, smoking status, age, MP presence in primary specimen, tumour focality, size, and CIS concomitant incidence. After matching, 342 patients were included in the analysis (Table 3).

Due to the fact that almost a half of patients were excluded after the PSM, the IPW was later performed for the whole population using the same variables for weighting as for the PSM adjusting (Figure 3).

Similarly, after PSM, neither RFS nor PFS differed for the five-year (Figure 4A,B) and total observation periods (data not shown). When IPW was implemented, borderline significance was observed for five-year RFS, with better results for the Post-BCG group (68.9% vs. 61.2%; *p* = 0.063) (Figure 4C). For other IPW survival analyses, differences in RFS and PFS did not reach statistical significance (Figure 4D).

## 4. Discussion

The issue of reTURB is recently gaining increased attention, with reasons both for and against this procedure. The EAU guidelines recommend the performance of reTURB a few weeks after the primary TURB procedure. However, in select urological centres, some patients undergo reTURB after the BCG induction. Reasons why reTURB is performed after BCG induction may include the diminished physical performance of patients after primary TURB and-even more important-a desire to speed up the commencement of BCG (a long waiting period for second, pre-BCG TURB). This might be particularly true during events like the COVID-19 pandemic.

Although this treatment is common, the issue of the performance of reTURB after the BCG induction has not been thoroughly analysed yet. In this study, a uniform group of patients with primary T1HG cancers was analysed. Out of a total of 645 patients identified from referral centres, 248 (38%) underwent reTURB after the BCG induction was completed. It was shown that the performance of reTURB after the induction course of BCG did not influence the oncological outcomes of treated patients. In a simple survival analysis, the results of RFS and PFS did not differ significantly between examined groups.

Furthermore, after PSM application, RFS and PFS did not differ significantly statistically for any observation period. However, when IPW was applied, the results showed statistical significance (*p* = 0.063) for the five-year observation of RFS.

Therapeutic decisions in the NMIBC setting are mainly based on the results of the initial TURB. However, primary TURB may unintentionally fail to control all present lesions or reliably detect cancer muscular infiltration.

Literature data show that up to half of patients are found to have residual disease after a completely resected T1 NMIBC. Additionally, some of them are finally proven to be muscle invasive disease [3,7,8]. It is also worth noting that patients with residual tumours in reTURB were shown to have higher risk of further recurrence and progression [9,10]. Because of the abovementioned TURB disadvantages, the reTURB concept was created in the mid-1990s [11]. The main aim of reTURB is to correctly assess the possible muscular invasion and resect all residual cancer. During the procedure, all gross lesions, as well as resection scars and/or oedematous/suspicious areas, are resected deep enough to include a sufficient amount of muscle in the histopathological specimens.

In 35% of reTURBs, some residual lesions were found. Interestingly, the rate of residual disease was lower in the Post-BCG group (38% vs. 30%; *p* = 0.029), possibly because of the BCG effect. Even though there is a statistically different occurrence of MP in primary specimens, the higher incidence rate of residual disease detected in reTURB was preserved after PSM, which was adjusted for MP (data not presented). Out of all cases of residual tumours, 105 (46%) were T1 tumours: 70 in the Pre-BCG group and 35 in Post-BCG. The difference was not statistically significant (*p* = 0.860). As stated in the available literature and as shown in this study, a viable tumour in reTUBR is associated with an increased risk of cancer recurrence and progression (2.6- and 3-fold higher, respectively). Unfortunately, the number of T2 lesions detected in reTURB was not reliably recorded in all centres, so these patients were not included in the analysis. Similarly, the influence of a time delay (resulted from T2 diagnosis at post-BCG reTURB) on some patients’ survival was not assessed.

Despite the advantages of the procedure, the influence of reTURB on oncological survival is not clear since the available studies present heterogenous populations treated with various adjuvant modalities. In a paper by Calo et al. that included 118 patients from a prospectively maintained nonrandomized high-risk NMIBC database, the authors showed that reTURB was not associated with any oncological benefit in RFS, PFS, and cancer-specific survival (CSS) [12]. In the largest study to date, performed by Gontero et al., reTURB had a positive impact on survival only if MP involvement was not present in the primary specimen [13]. Finally, in a recent meta-analysis of six studies including more than 3000 patients, it was shown that reTURB does not improve survival outcomes in patients with T1 bladder tumours [14].

The optimal timing of reTURB is still to be confirmed. In the literature, we found various time periods between primary and reTURB, ranging from a few days to 12 weeks [7]. Baltaci et al. showed that reTURB should not be performed more than 42 days after the primary TURB [15]. Krajewski et al. demonstrated that there is no benefit to reTURB if the procedure is performed eight weeks after the primary TURB and assumed that optimal timing of reTURB is from 2 to 6 weeks after initial TURB. Also, it was shown that each day of reTURB postponement results in a 4% increase in the risk of all analysed clinical events [16]. Because of this, it might be hypothesized that the time intervals in post-BCG reTURB play a similar role as in conventional reTURB. However, this aspect was not analysed in our study.

### Study Limitations

Although our study presented several strengths, it also had some limitations. First, the majority of the clinical and pathological data was collected retrospectively. However, when compared with recently published high-quality data, this study population is representative in terms of basic characteristics [17]. Recurrence and progression rates may seem low when compared with classic EORTC or CUETO nomograms, but our observations did not differ significantly from those in similar studies [18]. Additionally, allocation to groups was not intentionally randomized. Conclusively, to overcome the limitations of a retrospective design, we performed a PSM and IPW analysis matching patients for baseline characteristics. We did not include BCG strain in the matching analysis, as it led to a substantial limitation on case numbers and consequently a limitation on the number of adverse events. Secondly, one of the major post-BCG reTURB limitations is the fact that patients who are understaged during primary TURB will be diagnosed with a delay that could be fatal. Third, the data included in this study were mainly gathered from outpatient departments performing BCG procedures. This meant that thorough details about patients who were not qualified or not eligible for BCG (according to the total amount of T2 at reTURB in each centre, further treatment modalities of tumours that progressed, survival, etc.) were excluded from the analysis. Fourth, to preserve the homogeneity of the population, only patients who received at least five induction and two maintenance instillations, representing adequate BCG exposure, were included. Unfortunately, this might have resulted in the exclusion of some suitable patients (e.g., patients with a poor outcome at reTURB after BCG induction). Fifth, there was no central specimen review and no substaging of T1 tumours. Sixth, all data used in this paper originated from large oncological centres. Therefore, the results of this study may not be applicable to centres with less experience of bladder cancer treatment. Seventh, data regarding the experience of surgeons and technical details (e.g., en bloc, reTURB range) were not available and therefore were not included in the analysis. Also, data on the WHO 1973 grade, immediate single-dose chemotherapy, LVI, VH, and prostatic involvement of the tumours were not uniformly reported and/or unreliable and therefore were not included in the analysis. Finally, we did not perform CSS analysis, because the number of events (cancer-specific deaths) was low and therefore not statistically representative.

## 5. Conclusions

Despite a lack of evidence for its efficacy, the reTURB procedure is performed at many urological centres following the BCG induction course. Our results suggest that there might be no difference in recurrence-free survival or progression-free survival between patients with high-grade T1 disease who had reTURB before BCG induction and patients with reTURB after BCG induction. This study also confirms that patients with residual tumours in reTURB are characterised by lower survival rates compared with patients with negative reTURB.

## Figures and Tables

**Figure 1 jcm-09-03306-f001:**
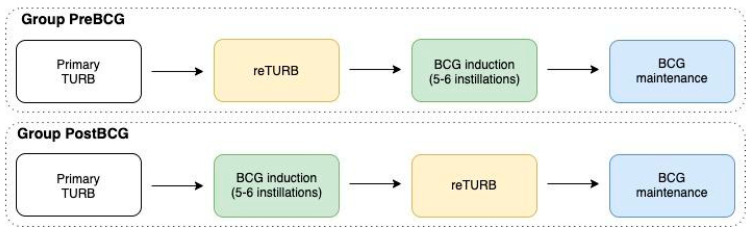
Therapeutic sequence after primary TURB (transurethral resection of bladder tumour).

**Figure 2 jcm-09-03306-f002:**
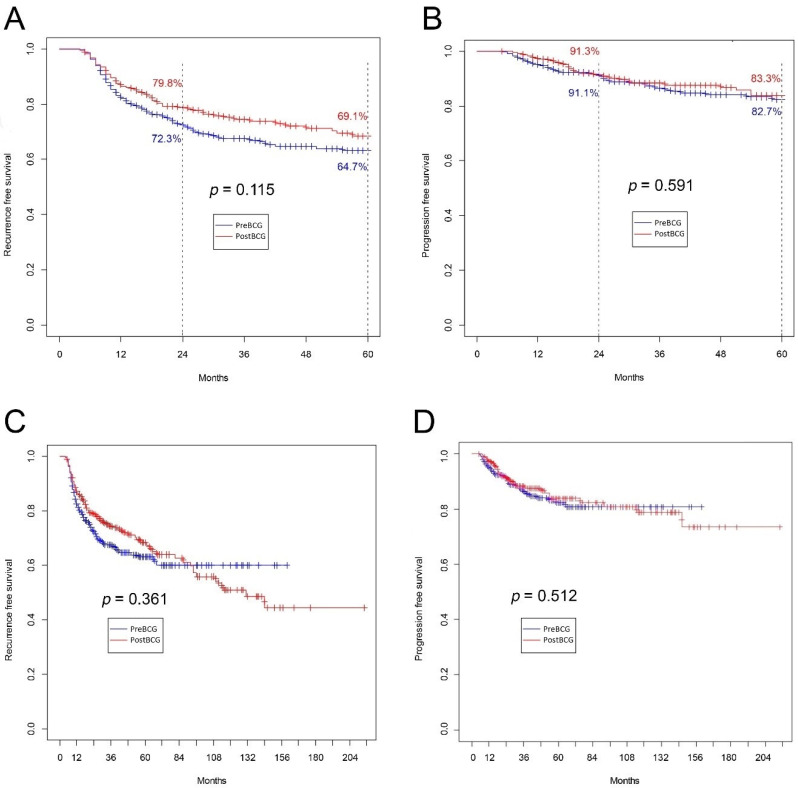
(**A**) Recurrence-free survival for five-year follow-up (*p* = 0.115); (**B**) progression-free survival for five-year follow-up (*p* = 0.591); (**C**) recurrence-free survival for whole observation period (*p* = 0.36); (**D**) progression-free survival for whole observation period (*p* = 0.512).

**Figure 3 jcm-09-03306-f003:**
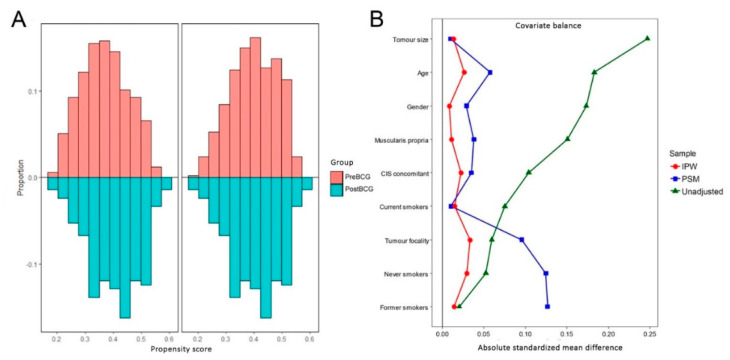
(**A**) Histogram distribution of propensity scores before and after PSM between the study groups. (**B**) Covariate balance before adjustment and after PSM and IPW (inverse probability of treatment weighting). IPW, inverse-probability weighting; PSM, propensity-score-matching.

**Figure 4 jcm-09-03306-f004:**
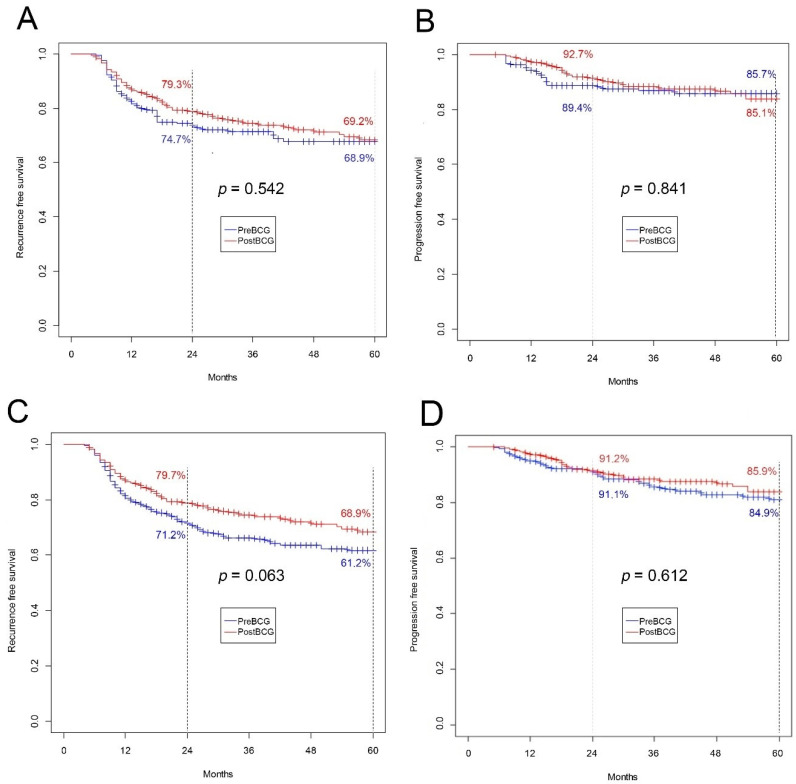
(**A**) Recurrence-free survival for five-year follow-up after PSM (*p* = 0.542); (**B**) progression-free survival for five-year follow-up after PSM (*p* = 0.841); (**C**) recurrence-free survival for five-year follow-up after IPW (*p* = 0.063); (**D**) progression-free survival for five-year follow-up after IPW (*p* = 0.612). IPW, inverse-probability weighting; PSM, propensity-score-matching.

**Table 1 jcm-09-03306-t001:** The patients’ baseline characteristics. Chi^2^ and Mann–Whitney test *p*-values of the differences between the two study groups.

	All Patients (*n* = 645)	Pre-BCG Group (*n* = 397; 61.6%)	Post-BCG Group (*n* = 248; 38.4%)	*p*-Value
Age (mean; SD)	66.5; 10.9	65.9; 11.8	67.2; 9.9	0.074
Gender (M/F)	538/107 (83.4/16.6%)	321/76 (80.9/19.1%)	217/31 (87.5/12.5%)	**0.027**
Smoking history; *n* (%)				0.293
Never	191 (29.6%)	121 (30.5%)	70 (28.2%)	
Former	266 (41.2%)	165 (41.6%)	101 (40.7%)	
Current	167 (25.9%)	95 (23.9%)	72 (29%)	
Unknown	21 (3.3%)	16 (4%)	5 (2%)	
Concomitant CIS; *n* (%)				0.054
Yes	126 (19.5%)	67 (16.9%)	59 (23.8%)	
No	514 (79.7%)	328 (82.6%)	186 (75%)	
unknown	5 (0.8%)	2 (0.5%)	3 (1.2%)	
Tumour size; *n* (%)				0.102
<3 cm	315 (48.8%)	187 (47.1%)	137 (55.2%)	
≥3 cm	279 (43.3%)	180 (45.3%)	94 (37.9%)	
Unknown	51 (7.9%)	30 (7.6%)	17 (6.9%)	
Tumour focality; *n* (%)				0.254
Solitary	299 (46.4%)	180 (45.3%)	119 (48%)	
Multiple	317 (49.2%)	195 (49.1%)	122 (49.2%)	
Unknown	29 (4.5%)	22 (5.5%)	7 (2.8%)	
Muscularis propria in the primary specimen				**0.034**
Yes	467 (72.4%)	279 (70.3%)	188 (75.8%)	
No	141 (21.9%)	99 (24.9%)	42 (16.9%)	
Unknown	37 (5.7%)	19 (4.8%)	18 (7.3%)	
Residual NMIBC at reTURB (yes/no)	226/419 (35/65%)	152/245 (38.3/61.7%)	74/174 (29.8/70.2%)	**0.029**
Stage of residual disease at reTURB				
T1	105 (46.5%)	70 (46.1%)	35 (47.3%)	0.860
Ta	66 (29.2%)	44 (28.9%)	22 (29.7%)	0.903
CIS	55 (24.3%)	38 (25%)	17 (23%)	0.739
Muscularis propria in the reTURB specimen (yes/no)	439/181/25 (68.1/28.1/3.9%)	268/115/14 (67.5/29/3.5%)	171/66/11 (69/26.6/4.4%)	0.714
BCG strain; *n* (%)				**0.017**
Moreau	138 (21.4%)	99 (24.9%)	39 (15.7%)	
TICE	272 (42.4%)	153 (38.5%)	119 (48%)	
RIVM	180 (27.9%)	114 (28.7%)	66 (26.6%)	
Other	55 (8.5%)	31 (7.8%)	24 (9.7%)	
Total number of BCG instillations (mean; SD)	14.7; 6.5	14.4; 6.2	14.9; 6.0 6.5	0.101
Observation time in months (median; IQR)	43.5 (24.5–58.5)	40 (27.6–58.6)	45.4 (23.3–58.0)	**>0.001**
Recurrence	218 (33.8%)	128 (32.2%)	90 (36.6%)	0.290
Progression	94 (14.6%)	54 (13.6%)	40 (16.1)	0.376
Cancer-specific death	38 (5.9%)	26 (6.5%)	12 (4.8%)	0.369

The value of adjusted *p* < 0.05 was considered statistically significant (in bold). Abbreviations: SD, standard deviation; IQR, interquartile range; M, male; F, female; CIS, carcinoma in situ; NMIBC, non-muscle-invasive bladder cancer

**Table 2 jcm-09-03306-t002:** Multivariable analysis assessing factors associated with disease recurrence and progression.

	Recurrence-Free Survival			Progression-Free Survival		
Variable	HR	95% CI	*p*-Value	HR	95% CI	*p*-Value
Age	1.00	0.99–1.02	0.537	1.00	0.98–1.02	0.829
Gender (**female** vs. male)	1.07	0.67–1.46	0.971	0.97	0.47–1.58	0.627
Smoking (never/**any**)	1.02	0.74–1.39	0.917	1.01	0.60–1.57	0.902
MP (**yes**/no)	0.92	0.64–1.32	0.657	0.76	0.47–1.31	0.360
Concomitant CIS (**yes** vs. no)	1.35	0.95–1.92	0.094	1.33	0.79–2.22	0.278
Size (≤3 cm vs. **>3 cm**)	1.05	0.72–1.30	0.802	1.26	0.79–2.01	0.332
Focality (solitary vs. **multiple**)	0.99	0.73–1.33	0.932	1.35	0.86–2.13	0.194
reTURB before/**after**	0.93	0.67–1.27	0.667	0.96	0.60–1.53	0.864
Residual tumour at reTURB **yes**/no	**2.62**	**1.94–3.54**	**0.000**	**3.01**	**1.90–4.77**	**0.000**

The value of adjusted *p* < 0.05 was considered statistically significant (in bold); HR > 1 worse outcome for the option in bold, HR < 1 better outcome for the option in bold. Abbreviations: HR, hazard ratio; 95% CI, 95% confidence interval; CIS, carcinoma in situ; MP, muscularis propria.

**Table 3 jcm-09-03306-t003:** The patients’ baseline characteristics after propensity-score-matched analysis (PSM). Chi^2^ and Mann–Whitney test *p*-values of the differences between the two study groups.

	All Patients (*n* = 645)	Pre-BCG Group after Matching (*n* = 171)	Post-BCG Group after Matching (*n* = 171)	*p*-Value
Age (mean; SD)	66.5; 10.9	66.2; 10.9	67.6; 9.9	0.248
Gender M/F; (%)	538/107 (83.4/16.6%)	146/25 (85.4/14.6%)	150/21 (87.7/12.3%)	0.526
Smoking history; *n* (%)				0.931
Never	191 (29.6%)	48 (28.1%)	51 (29.8%)	
Former	266 (41.2%)	73 (42.7%)	69 (40.4%)	
Current	167 (25.9%)	47 (27.5%)	49 (28.6%)	
Unknown	21 (3.3%)	3 (1.7%)	2 (1.2%)	
Concomitant CIS; *n* (%)				0.634
Yes	126 (19.5%)	30 (17.5%)	37 (21.6%)	
No	514 (79.7%)	140 (81.9%)	133 (77.8%)	
unknown	5 (0.8%)	1 (0.6%)	1 (0.6%)	
Tumour size; *n* (%)				0.393
<3 cm	315 (48.8%)	80 (46.8%)	68 (39.8%)	
≥3 cm	279 (43.3%)	82 (47.9%)	91 (53.2%)	
Unknown	51 (7.9%)	9 (5.3%)	12 (7%)	
Tumour focality; *n* (%)				0.267
Solitary	299 (46.4%)	88 (51.5%)	83(48.5%)	
Multiple	317 (49.2%)	77 (45%)	86 (50.3%)	
Unknown	29 (4.5%)	6 (3.5%)	2 (1.2%)	
Muscularis propria in the primary specimen				0.949
Yes	467 (72.4%)	140 (81.9%)	142 (83%)	
No	141 (21.9%)	23 (13.4%)	22 (12.9%)	
Unknown	37 (5.7%)	8 (4.7%)	7 (4.1%)	

The value of adjusted *p* < 0.05 was considered statistically significant. Abbreviations: SD, standard deviation; M, male; F, female; CIS, carcinoma in situ.

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
