# Peer review of "Restaging Transurethral Resection of Bladder Tumours after BCG Immunotherapy Induction in Patients with T1 Non-Muscle-Invasive Bladder Cancer Might not Be Associated with Oncologic Benefit"

_jcm, 2020, doi:10.3390/jcm9103306_

Round 1
Reviewer 1 Report
Major comments:
- On line 152 (result section) authors refer that the groups did not differ statistically in tumor size. But if you take a closer look at Table 1 where the tumor size is being referred there is a difference in tumor size between the 2 groups. 47.1% vs 55.2% for tumors <3cm amd 45.3% vs 37.9% for tumors >=3cm between pre-BCG group and post-BCG group. It's not a huge difference but for sure you cannot concider them the same.
- On the same concept, on lines 162-163 authors mention that "no differences were found for the whole observation period, either for RFS figure 2C or PFS figure 2D". But based on figure 2C around months 140(~11-12 years) there is a difference in RFS percentage ~60% versus ~43%. Do not authors consider this as a difference? For the 5 year RFS indeed there is no significant difference, but for the whole observation period there is for RFS.
- On lines 190 and 191 authors mention "....for the five-year and total observation periods (figure 4A,B)". But in figure 4A and B we can see only the 5-year and not the whole time RFS and PFS. It would be important to add these graphs after the PSM.
- In general based on the statistical analysis and the p-values which in most of the cases are very high, I wouldn't say with confidence that there is no difference in RFS and PFS rates, regardless of whether patients have reTURB performed before or after BCG induction, but based on your analysis I would rather say that this is not clear. Based on the high p-values that reflect most likely the small sample size and the variability of the data and as you very nicely mention the limitations of this study at the "Study limitations" part, I would recommend to rephrase the title and the parts that you strongly state that there is no oncologic benefit whether you perform the BCG induction before or after the reTURB.
Minor comments:
- Rephrase the sentence "Viable tumours....as a recurrence" on lines 113-114.
- Under Table 1, line 149 add the IQR that is interquartile range.
- Similarly, line 183 under Table 3 add the SD.
- Lines 232-234 "Out of all cases...(p=0.239; data not presented)". But the data are presented in Table 1 and the p value is not 0.239 but 0.860
- Line 241. "in a paper by Calo et al. ..." which paper is that? Is not included in the references here.
- Line 249 "Baltici et al..." is Baltaci et al.
- Line 250 you should correct it to "reTURB should not be performed more than 42 days after the primary TURB"
- Line 251-252. It would be nice to refer what is the optimal they recommend in this paper (which is also your paper). They refer as optimal 2-6 weeks after primary reTURB.
- Line 254. Authors refer group B. Which is group B and also where is group A? Explain this better.
- Line 282, explain what CSS is in parenthesis (Cancer specific survival)
Author Response
Major comments:
- On line 152 (result section) authors refer that the groups did not differ statistically in tumor size. But if you take a closer look at Table 1 where the tumor size is being referred there is a difference in tumor size between the 2 groups. 47.1% vs 55.2% for tumors <3cm amd 45.3% vs 37.9% for tumors >=3cm between pre-BCG group and post-BCG group. It's not a huge difference but for sure you cannot consider them the same.
Our response: All statistical analyses were performed by experienced statistician, who reviewed and confirmed the consistency of all calculations once more. According to results of Chi2 test, the difference in tumor size between groups presented in Table 1 is statistically insignificant, with p-value >0.05 (p=0.102). It has to be emphasized, that statistical difference is not only based on subjective incomparability of percentages, but also on other factors included in the mathematical model of Chi2 test.
It is important to note that also unknown cases were included in the analysis. If our justification is insufficient we can provide raw data for verification.
Notwithstanding, because of some subtle differences (e.g. in tumor size - p-value 0.102 can be considered as slight trend towards statistical significance), and some more evident and statistically significant differences (e.g. muscularis propria presence) the matching methods (propensity score matching - PSM, and inverse probability weighting - IPW) were implemented. The Figure 3B shows very clearly the effect of matching of each variable, including tumor size.
- On the same concept, on lines 162-163 authors mention that "no differences were found for the whole observation period, either for RFS figure 2C or PFS figure 2D". But based on figure 2C around months 140(~11-12 years) there is a difference in RFS percentage ~60% versus ~43%. Do not authors consider this as a difference? For the 5 year RFS indeed there is no significant difference, but for the whole observation period there is for RFS.
Our response: Once more, all statistical analyses were performed by experienced statistician, who reviewed and confirmed the consistency of all calculations again. The difference of recurrence-free survival for whole observation period is clearly not statistically significant (p = 0.36). The final result of a log-rank test is based on solid mathematical model. If some subjective visual differences exist for highly selected and limited time points/periods (especially for the least reliable far right - smallest cases number) it does not necessarily mean that the difference for the whole time period is statistically significant. Again, if our justification is insufficient we can provide raw data for verification.
- On lines 190 and 191 authors mention "....for the five-year and total observation periods (figure 4A,B)". But in figure 4A and B we can see only the 5-year and not the whole time RFS and PFS. It would be important to add these graphs after the PSM.
Our response: Figures representing total observation period after both PSM and IPW do not provide any statistically significant and important information. They are relatively comparable to K-M curves before matching (Figure 2C,D). Therefore, those additional 4 graphs were not included in the manuscript to avoid quantity chaos, and we stated it as “data not shown” (line 203-204). However, if the Reviewer finds it necessary, we can provide a supplementary figure representing total observation period after PSM and IPW.
- I would recommend to rephrase the title and the parts that you strongly state that there is no oncologic benefit whether you perform the BCG induction before or after the reTURB.
Our response: The title, abstract and conclusions has been corrected accordingly to reviewers suggestion (line 2-5; 65-67; 300-302).
Minor comments:
- Rephrase the sentence "Viable tumours....as a recurrence" on lines 113-114.
Our response: The sentence was corrected (line 121-122)
- Under Table 1, line 149 add the IQR that is interquartile range.
Our response: The abbreviation was added (line 158)
- Similarly, line 183 under Table 3 add the SD.
Our response: The abbreviation was added (line 194)
- Lines 232-234 "Out of all cases...(p=0.239; data not presented)". But the data are presented in Table 1 and the p value is not 0.239 but 0.860
Our response: This mistake resulted from our inattention and it was corrected (line 246)
- Line 241. "in a paper by Calo et al. ..." which paper is that? Is not included in the references here.
Our response: Reference to this paper was added (line 254-256 and 364-366)
- Line 249 "Baltici et al..." is Baltaci et al
Our response: Author’s name was corrected (line 262)
- Line 250 you should correct it to "reTURB should not be performed more than 42 days after the primary TURB"
Our response: The sentence was corrected as suggested (line 263)
- Line 251-252. It would be nice to refer what is the optimal they recommend in this paper (which is also your paper). They refer as optimal 2-6 weeks after primary reTURB.
Our response: Information was added as suggested (line 263-266).
- Line 254. Authors refer group B. Which is group B and also where is group A? Explain this better.
Our response: The sentence was corrected as it referred to post-BCG reTURB (line 268)
- Line 282, explain what CSS is in parenthesis (Cancer specific survival)
Our response: We stated that analysis of cancer specific survival (CSS) was not performed because number of events (cancer-specific deaths - data is available in Table 1) was low and therefore not statistically representative (line 295-297)

Reviewer 2 Report
The authors completely ignore the fact that the re-TURB is mainly to ensure that no residual tumour is left and the importance to identify muscleinvasive tumours. The study excluded the T2+ tumours from analysis, which in my opinion makes this paper quite interesting. In my country, or other neighbouring countries, I have never encountered the phenomenon of giving BCG before reresection. Some data were missing which would be interesting to know why they were missing, e g single instillation, LVI or variant histology.
Author Response
- The authors completely ignore the fact that the re-TURB is mainly to ensure that no residual tumour is left and the importance to identify muscle-invasive tumours. The study excluded the T2+ tumours from analysis, which in my opinion makes this paper quite interesting.
Our response: This study does not ignore T2 patients, but excludes them electively. By definition, T2 tumours are not qualified for BCG, but undergo radical (or any other) treatment. It is vital to acknowledge that this study does not aim to present epidemiological data of TURB understaging, and does not present whole population of patients treated with TURB, reTURB and BCG in given centre. Therefore, only patients that were qualified and treated with BCG are included, which means, no T2 tumors are shown (neither in the primary TURB nor in the reTURB). Those patients were disqualified from conservative treatment and were generally treated radically. As the main aim of the study was to compare various models of conservative therapy for only T1HG (which is not suitable for T2 tumors) and not to provide epidemiological data of all patients treated in all centres, those cases were not included. This is underlined in the various section of the paper.
- In my country, or other neighbouring countries, I have never encountered the phenomenon of giving BCG before reresection.
Our response: We completely agree with the reviewer that BCG, which is given before reTURB, is not universally performed. However, it is performed in several high volume urological referral centres across the Europe. Therefore, the hypothesis that this protocol is not worse than the classic one was formed and this analysis was performed as the result.
3. Some data were missing which would be interesting to know why they were missing, eg. single instillation, LVI or variant histology.
Our response: We highlighted in the limitations section (line 294-295) that LVI, VH, and prostatic involvement of the tumours were not uniformly reported in several participating centres and therefore not included in the analysis. Yet, this issue is not possible to be reliably assessed in retrospective form. When initial patients were treated, some of these factors were not known or not considered to be important. Moreover, LVI and VH are very difficult to assessment in TURB specimens, especially in small and fragmented tumors.

Round 2
Reviewer 2 Report
In Table 1, under "Stage of residual disease" (n=645) only 226 results are shown. I assume that the misssing 419 patients had T0 or T2? It would be interesting to know. I do not Think that Figure 3 adds any valuable information.
As stated, the study has several limitations. The conclusion has been altered accordingly.
Author Response
- In Table 1, under "Stage of residual disease" (n=645) only 226 results are shown. I assume that the misssing 419 patients had T0 or T2? It would be interesting to know.
Our response: The residual disease in Table 1 consist only NMIBCs – this is T1, Ta and CIS. 226 (35%) patients had residual disease (NMIBC) and 419 (65%) patients had no residual disease. As stated in Material and Methods section, T2 tumors are not included in any analysis of the study. The table description has been changed accordingly (page 4, Table 1, yellow mark).
- I do not Think that Figure 3 adds any valuable information.
Our response: Figure 3 shows the changes of propensity scores and covariate balance before adjustment and after implementation of matching techniques (propensity score matching [PSM] and inverse probability weighting [IPW]). Therefore, reduction in propensity scores and standardized mean difference (SMD) for all covariates is clearly presented - as an evidence that the population in Pre- and Post-BCG group is uniform in terms of primary variables. Presentation of matching results in Figure (Figure 3) and Table (Table 3) was required by another Reviewer. However, if the Reviewer consider the Figure 3 unnecessary, we can remove it from manuscript or provide it in supplementary form.
